# TWO STEPS AT A TIME — TAKING GAN TRAINING IN STRIDE WITH TSENG'S METHOD

## ABSTRACT

Motivated by the training of Generative Adversarial Networks (GANs), we study methods for solving minimax problems with additional nonsmooth regularizers. We do so by employing *monotone operator* theory, in particular the *Forward-Backward-Forward (FBF)* method, which avoids the known issue of limit cycling by correcting each update by a second gradient evaluation. Furthermore, we propose a seemingly new scheme which recycles old gradients to mitigate the additional computational cost. In doing so we rediscover a known method, related to *Optimistic Gradient Descent Ascent (OGDA)*. For both schemes we prove novel convergence rates for convex-concave minimax problems via a unifying approach. The derived error bounds are in terms of the gap function for the ergodic iterates. For the deterministic and the stochastic problem we show a convergence rate of $\mathcal{O}(1/k)$ and $\mathcal{O}(1/\sqrt{k})$, respectively. We complement our theoretical results with empirical improvements in the training of Wasserstein GANs on the CIFAR10 dataset.

## 1 INTRODUCTION

*Generative Adversarial Networks (GANs)* (Goodfellow et al., 2014) have proven to be a powerful class of generative models, producing for example unseen realistic images. Two neural networks, called generator and discriminator, compete against each other in a game. In the special case of a zero sum game this task can be formulated as a minimax (aka saddle point) problem.

Conventionally, GANs are trained using variants of (stochastic) *Gradient Descent Ascent (GDA)* which are known to exhibit oscillatory behavior and thus fail to converge even for simple bilinear saddle point problems, see Goodfellow (2016). We therefore propose the use of methods with provable convergence guarantees for (stochastic) convex-concave minimax problems, even though GANs are well known to not warrant these properties. Along similar considerations an adaptation of the *Extragradient method (EG)* (Korpelevich, 1976) for the training of GANs was suggested in Gidel et al. (2019), whereas Daskalakis et al. (2018); Daskalakis & Panageas (2018); Liang & Stokes (2019) studied *Optimistic Gradient Descent Ascent (OGDA)* based on *optimistic mirror descent* (Rakhlin & Sridharan, 2013a;b). We however investigate the *Forward-Backward-Forward (FBF)* method (Tseng, 1991) from monotone operator theory, which uses two gradient evaluations per update, similar to EG, in order to circumvent the aforementioned issues.

Instead of trying to improve GAN performance via new architectures, loss functions, etc., we contribute to the theoretical foundation of their training from the point of view of optimization.

**Contribution.** Establishing the connection between GAN training and *monotone inclusions* motivates to use the FBF method, originally designed to solve this type of problems. This approach allows to naturally extend the constrained setting to a regularized one making use of the proximal operator.

We also propose a variant of FBF reusing previous gradients to reduce the computational cost per iteration, which turns out to be a known method, related to OGDA. By developing a unifying scheme that captures FBF and a generalization of OGDA, we reveal a hitherto unknown connection. Using this approach we prove novel non asymptotic convergence statements in terms of the minimax gap for both methods in the context of saddle point problems. In the deterministic and stochastic setting we obtain rates of $\mathcal{O}(1/k)$ and $\mathcal{O}(1/\sqrt{k})$, respectively. Concluding, we highlight the relevance of

our proposed method as well as the role of regularizers by showing empirical improvements in the training of Wasserstein GANs on the CIFAR10 dataset.

**Organization.** This paper is structured as follows. In Section 2 we highlight the connection of GAN training and monotone inclusions and give an extensive review of methods with convergence guarantees for the latter. The main results as well as a precise definition of the measure of optimality are discussed in Section 3. Concluding, Section 4 illustrates the empirical performance in the training of GANs as well as solving bilinear problems.

## 2 GAN TRAINING AS MONOTONE INCLUSION

The GAN objective was originally cast as a two-player zero-sum game between the discriminator $D_y$ and the generator $G_x$ (Goodfellow et al., 2014) given by

$$\min_x \max_y \ \mathbb{E}_{\rho \sim q}[\log(D_y(\rho))] + \mathbb{E}_{\zeta \sim p}[\log(1 - D_y(G_x(\zeta)))],$$

exhibiting the aforementioned minimax structure. Due to problems with vanishing gradients in the training of such models, a successful alternative formulation called *Wasserstein GAN (WGAN)* (Arjovsky et al., 2017) has been proposed. In this case the minimization tries to reduce the Wasserstein distance between the true distribution $q$ and the one learned by the generator. Reformulating this distance via the Kantorovich Rubinstein duality leads to an inner maximization over 1-Lipschitz functions which are approximated via neural networks, yielding the saddle point problem

$$\min_x \max_{y : \|D_y\|_{\text{Lip}} \leq 1} \mathbb{E}_{\rho \sim q}[D_y(\rho)] - \mathbb{E}_{\zeta \sim p}[D_y(G_x(\zeta))].$$

### 2.1 CONVEX-CONCAVE MINIMAX PROBLEMS

Due to the observations made in the previous paragraph we study the following abstract minimax problem

$$\min_{x \in \mathbb{R}^d} \max_{y \in \mathbb{R}^n} \ \Psi(x,y) := f(x) + \mathbb{E}_{\xi \sim Q}\left[\Phi(x,y;\xi)\right] - h(y), \tag{1}$$

where the convex-concave coupling function $\Phi(x,y) := \mathbb{E}_{\xi \sim Q}\left[\Phi(x,y;\xi)\right]$, which hides the stochasticity for ease of notation, is differentiable with $L$-Lipschitz continuous gradient. The proper, convex and lower semicontinuous functions $f : \mathbb{R}^d \to \mathbb{R} \cup \{+\infty\}$ and $h : \mathbb{R}^n \to \mathbb{R} \cup \{+\infty\}$ act as regularizers. A solution of (1) is given by a so-called *saddle point* $(x^*, y^*)$ fulfilling for all $x$ and $y$

$$\Psi(x^*, y) \leq \Psi(x^*, y^*) \leq \Psi(x, y^*).$$

In the context of two-player games this corresponds to a pair of strategies, where no player can be better off by changing just their own strategy.

For the purpose of this motivating section, we will restrict ourselves for now to the special case of the *deterministic constrained* version of (1), given by

$$\min_{x \in X} \max_{y \in Y} \ \Phi(x,y),$$

where $f$ and $h$ are given by indicator functions of closed convex sets $X$ and $Y$, respectively. The indicator function $\delta_C$ of a set $C$ is defined as $\delta_C(z) = 0$ for $z \in C$ and $\delta_C(z) = +\infty$ otherwise.

### 2.2 MINIMAX PROBLEMS AS MONOTONE INCLUSIONS

If the coupling function $\Phi$ is convex-concave and differentiable then solving (1) is equivalent to solving the first order optimality conditions which can be written as a so-called *monotone inclusion* with $w = (x,y) \in \mathbb{R}^m$ and $m = d + n$, given by

$$0 \in F(w) + N_\Omega(w). \tag{2}$$

The entities involved are

$$F(x,y) := (\nabla_x \Phi(x,y), -\nabla_y \Phi(x,y)), \tag{3}$$

and the *normal cone* $N_\Omega$ of the convex set $\Omega := X \times Y$. The normal cone mapping is given by

$$N_\Omega(w) = \{v \in \mathbb{R}^m : \langle v, w' - w \rangle \leq 0 \quad \forall w' \in \Omega\},$$

for $w \in \Omega$ and $N_\Omega(w) = \emptyset$ for $w \notin \Omega$. Here, the operators $F$ and $N_\Omega$ satisfy well known properties from convex analysis (Bauschke & Combettes, 2011), in particular the first one is monotone (and Lipschitz if $\nabla \Phi$ is so) whereas the latter one is maximal monotone. We call a, possibly *set-valued*, operator $A$ from $\mathbb{R}^m$ to itself monotone if

$$\langle u - u', z - z' \rangle \geq 0 \quad \forall u \in A(z), u' \in A(z').$$

We say $A$ is maximal monotone, if there exists no monotone operator $A'$ such that the graph of $A$ is properly contained in the graph of $A'$.

Problems of type (2) have been studied thoroughly in convex optimization, with the most established solution methods being *Extragradient* (Korpelevich, 1976) and *Forward-Backward-Forward* (Tseng, 1991). Both methods are known to generate sequences of iterates converging to a solution of (2). Note that in the unconstrained setting (i.e. if $\Omega$ is the entire space) both of these algorithms even produce the same iterates.

### 2.3 Solving monotone inclusions

The connection between monotone inclusions and saddle point problems is of course not new. The application of Extragradient (EG) to minimax problems has been studied in the seminal paper Nemirovski (2004) under the name of *Mirror Prox* and a convergence rate of $\mathcal{O}(1/k)$ in terms of the function values has been proven. Even a stochastic version of the Mirror Prox algorithm has been studied in Juditsky et al. (2011) with a convergence rate of $\mathcal{O}(1/\sqrt{k})$. Applied to problem (2), with $P_\Omega$ being the projection onto $\Omega$, it iterates

$$\text{EG:} \left| \begin{array}{l} w_k = P_\Omega[z_k - \alpha_k F(z_k)] \\ z_{k+1} = P_\Omega[z_k - \alpha_k F(w_k)]. \end{array} \right.$$

The Forward-Backward-Forward (FBF) method, introduced in Tseng (1991), has not been studied rigorously for minimax problems in terms of function values yet, despite promising applications in Boţ et al. (2020) and *its advantage of it only requiring one projection*, whereas EG needs two. It is given by

$$\text{FBF:} \left| \begin{array}{l} w_k = P_\Omega[z_k - \alpha_k F(z_k)] \\ z_{k+1} = w_k + \alpha_k(F(z_k) - F(w_k)). \end{array} \right. \tag{4}$$

Both, EG and FBF, have the "disadvantage" of needing two gradient evaluations per iteration. A possible remedy — suggested in Gidel et al. (2019) for EG under the name of *extrapolation from the past* — is to recycle previous gradients. In a similar fashion we consider

$$\text{FBFp:} \left| \begin{array}{l} w_k = P_\Omega[z_k - \alpha_k F(w_{k-1})] \\ z_{k+1} = w_k + \alpha_k(F(w_{k-1}) - F(w_k)), \end{array} \right. \tag{5}$$

where we replaced $F(z_k)$ by $F(w_{k-1})$ twice in (4). As a matter of fact, the above method can be written exclusively in terms of the first variable $w_k$ by incrementing the index $k$ in the first update and then substituting in the second line. This results in

$$w_{k+1} = P_\Omega\left[w_k - \alpha_{k+1}F(w_k) + \alpha_k(F(w_{k-1}) - F(w_k))\right]. \tag{6}$$

This way we rediscover a known method which was studied in Malitsky & Tam (2020) for general monotone inclusions under the name of *forward-reflected-backward*. It reduces to *optimistic mirror descent* (Rakhlin & Sridharan, 2013a;b) in the unconstrained case with constant step size $\alpha_k = \alpha$, giving

$$w_{k+1} = w_k - \alpha(2F(w_k) - F(w_{k-1})) \tag{7}$$

which has been proposed for the training of GANs under the name of *Optimistic Gradient Descent Ascent (OGDA)*, see Daskalakis et al. (2018); Daskalakis & Panageas (2018); Liang & Stokes (2019).

All of the above methods and extensions rely solely on the monotone operator formulation of the saddle point problem where the two components $x$ and $y$ play a symmetric role. Taking the special minimax structure into consideration, Hamedani & Aybat (2018) showed convergence of a method

that uses an optimistic step (7) in one component and a regular gradient step in the other, thus requiring less storing of past gradients in comparison to (6).

On the downside, however, by reducing the number of required gradient evaluations per iteration, the largest possible step size is reduced from $1/L$ (see Korpelevich (1976) or Section 3) to $1/2L$ (see Gidel et al. (2019); Malitsky & Tam (2020); Malitsky (2015) or Section 3). To summarize, the number of required gradient evaluations is halved, but so is the step size, resulting in no clear net gain.

### 2.4 REGULARIZERS

The role of regularizers is well studied in many fields such as statistics (Tibshirani, 1996), signal processing (Palomar & Eldar, 2010) or inverse problems (Rudin et al., 1992). They serve different purposes such as inducing sparsity in the solution or conditioning of the problem. In the context of deep learning this has been explored from different perspectives, e.g. in incremental convex neural networks where neurons with zero weights are removed from the network and new ones are inserted according to different policies, see Bach (2017); Bengio et al. (2006); Rosset et al. (2007); Pieper & Petrosyan (2020). Other examples include the box-constraints for WGANs with weight clipping (see Arjovsky et al. (2017)) or spectral normalization (see Miyato et al. (2018)) which has so far rather been considered as part of the architecture, but can at the same time seen as a regularization term of the function values.

In the framework of monotone operator theory the optimality condition of the regularized minimax problem (1) can be written as

$$0 \in F(w) + \partial r(w), \tag{8}$$

where $r$ is given by $(x, y) \mapsto f(x) + h(y)$. The possibly set-valued operator $\partial r$ denotes the subdifferential of $r$ and is given by

$$\partial r(w) := \{v \in \mathbb{R}^m : \langle v, w' - w \rangle + r(w) \leq r(w') \quad \forall w' \in \mathbb{R}^m\}.$$

The monotone inclusion (8) generalizes (2) in a natural way, since $N_\Omega = \partial \delta_\Omega$. Similarly, the projection constitutes a special case of the so-called *proximal mapping* which for the function $r$ and $\lambda > 0$ is given by

$$\mathrm{prox}_{\lambda r}(w) := \underset{w' \in \mathbb{R}^m}{\arg\min} \left\{ r(w') + \frac{1}{2\lambda} \|w' - w\|^2 \right\}.$$

In particular, the proximal mapping of the indicator $\delta_\Omega$ yields the projection onto the set $\Omega$, i.e. $\mathrm{prox}_{\lambda \delta_\Omega} = P_\Omega$.

## 3 MAIN RESULTS

Motivated by the considerations above we study the inclusion problem

$$0 \in F(w) + \partial r(w), \tag{9}$$

where $F : \mathbb{R}^m \to \mathbb{R}^m$ is a monotone and Lipschitz operator and $r : \mathbb{R}^m \to \mathbb{R} \cup \{+\infty\}$ is a proper convex lower semicontinuous function.

### 3.1 MEASURE OF OPTIMALITY

There are two common quantities measuring the quality of a point with respect to the monotone inclusion (8). The most natural one is the distance to the solution set for which typically only asymptotic convergence can be proved. If $F$ arises from a saddle point problem (1) meaning that $F$ has the form (3), we want to use a more problem specific measure, the *minimax gap*, which for a point $w = (u, v) \in \mathbb{R}^d \times \mathbb{R}^n$ is given by

$$\sup_{y \in \mathbb{R}^n} \Psi(u, y) - \inf_{x \in \mathbb{R}^d} \Psi(x, v) \left( = \sup_{x \in \mathbb{R}^d, y \in \mathbb{R}^n} \Psi(u, y) - \Psi(x, v) \right). \tag{10}$$

This minimax gap can be interpreted from a game theoretic standpoint as the sum of the maximal payoffs achievable by the two players by playing their respective best responses, given the current

strategy of the opponent. In the more general monotone inclusion setting where no function values are available, an appropriate generalization of (10) is given for any $w \in \mathbb{R}^m$ by

$$\sup_{z \in \mathbb{R}^m} \langle F(z), w - z \rangle + r(w) - r(z).$$

If $r$ is the indicator $\delta_\Omega$ of the compact and convex set $\Omega$ it is clear that the supremum is only taken over $z \in \Omega$ and will thus be finite.

**The restricted gap.**   Since the problem (9) is in general unconstrained and the supremum can be infinite we consider instead, as done for example in Nesterov (2007), the *restricted* gap where the above supremum is taken over an auxiliary compact set $B \subset \mathbb{R}^m$ instead of the entire space. Note that the restricted gap is in general only a reasonable measure of optimality for elements of $B$. It is nonnegative on $B$ and zero for points of $B$ which solve (9). Additionally we want to be able to conclude that if a point $w^*$ has zero gap it solves (9). This is for example the case if $w^*$ is in the interior of $B$, which can always be ensured if $B$ is chosen large enough.

In order to capture both at the same time we define the following unifying gap

$$G_B(w) := \begin{cases} \sup_{(x,y) \in B} \Psi(u, y) - \Psi(x, v) & \text{if } F \text{ and } r \text{ come from (1)} \\ \sup_{z \in B} \langle F(z), w - z \rangle + r(w) - r(z) & \text{otherwise.} \end{cases} \tag{11}$$

## 3.2   METHODS

We now present a novel unifying scheme for solving problem (9), which generalizes FBF (4) and in addition recovers the method motivated in (5) as FBFp. Let us point out again that the latter algorithm was already introduced in Malitsky & Tam (2020) and corresponds to OGDA (Rakhlin & Sridharan, 2013a; Daskalakis et al., 2018; Daskalakis & Panageas, 2018) if $F$ stems from the minimax setting (3).

**Algorithm 3.1** (generalized FBF).   *For a starting point $z_0 \in \mathbb{R}^m$ and step sizes $\alpha_k > 0$ we consider for all $k \geq 0$*

$$\left| \begin{array}{l} w_k = \text{prox}_{\alpha_k r} (z_k - \alpha_k F(\Diamond_k)) \\ z_{k+1} = w_k + \alpha_k (F(\Diamond_k) - F(w_k)). \end{array} \right.$$

*For $\Diamond_k = z_k$ this reduces to the well known FBF method, whereas $\Diamond_k = w_{k-1}$, with the additional initial condition $w_{-1} = z_0$, recycles previous gradients (FBFp).*

Consider the scenario where $F$ is given as an expectation $\mathbb{E}_\xi[F(\cdot\,; \xi)]$, e.g. coming from (1), and only a stochastic estimator $F(\cdot\,; \xi)$ is accessible instead of $F$ itself. In this case we adapt Algorithm 3.1 in the following way.

**Algorithm 3.2** (generalized stochastic FBF).   *For a starting point $z_0 \in \mathbb{R}^m$ and step sizes $\alpha_k > 0$ we consider for all $k \geq 0$*

$$\left| \begin{array}{l} \xi_k \sim Q \quad (\text{optionally } \eta_k \sim Q) \\ w_k = \text{prox}_{\alpha_k r} (z_k - \alpha_k F(\Diamond_k; \triangle_k)) \\ z_{k+1} = w_k + \alpha_k (F(\Diamond_k; \triangle_k) - F(w_k; \xi_k)). \end{array} \right.$$

*For $\Diamond_k = z_k$ and $\triangle_k = \eta_k$ this results in a stochastic version of FBF, whereas $\Diamond_k = w_{k-1}$ and $\triangle_k = \xi_{k-1}$ recycles previous gradients (stochastic FBFp) with the additional initial condition $w_{-1} = z_0$ and $\xi_{-1} = \eta_0$.*

Even though both methods encompassed by the unifying scheme Algorithm 3.1 have been studied in the deterministic setting before, the stated convergence results are new. Note that while the rate for FBF is completely new our result for FBFp provides only a generalization of the known rate for OGDA, see Mokhtari et al. (2019). Similarly, the stochastic version of FBF has been considered before in Bot et al. (2019) and rates have been obtained, but only in terms of the fixed point residual and not the function values. However, we want to point out that the stochastic version of FBFp has not been considered prior to this work.

### 3.3 CONVERGENCE

Let in the following $B \subset \mathbb{R}^m$ be the compact set of the restricted (unifying) gap function (11) with $D := \sup_{w,z \in B} \|z - w\|$ denoting its diameter. For convenience in the estimation we assume that the starting point $z_0$ of the discussed methods is in $B$.

**Theorem 3.1** (deterministic). *Let $(w_k)_{k \geq 0}$ be the sequence generated by Algorithm 3.1. If*

   *(i)  FBF, i.e. $\Diamond_k = z_k$, with step size $\alpha_k = \alpha \leq 1/L$, or*

   *(ii)  FBFp, i.e. $\Diamond_k = w_{k-1}$, with step size $\alpha_k = \alpha \leq 1/2L$*

*is chosen, then for all $K \geq 1$ the averaged iterates $\bar{w}_K := \frac{1}{K} \sum_{k=0}^{K-1} w_k$ fulfill*

$$G_B(\bar{w}_K) \leq \frac{D^2}{2\alpha K},$$

*where $G_B$ is the restricted gap defined in (11).*

In order to derive similar convergence statements for the stochastic algorithm we need to assume (standard) properties of the gradient estimator $F(\cdot\,;\xi)$.

**Assumption 1.** *Unbiasedness: $\mathbb{E}_\xi[F(w;\xi)] = F(w) \, \forall w \in \mathbb{R}^m$.*

**Assumption 2.** *Bounded variance: $\mathbb{E}_\xi[\|F(w;\xi) - F(w)\|^2] \leq \sigma^2 \, \forall w \in \mathbb{R}^m$.*

In particular we actually only need the above assumption to hold for all iterates $w_k$. Such an hypothesis is in practice difficult to check, but could be exploited in special cases where additional properties of the variance and boundedness of the iterates are known a priori.

**Assumption 3.** *The samples $\xi_k$ are independent of the iterates $w_k$, for all $k \geq 0$.*

Equipped with these assumptions we are now able to prove the statement.

**Theorem 3.2** (stochastic). *Let Assumption 1, 2 and 3 hold and let $(w_k)_{k \geq 0}$ be the sequence generated by Algorithm 3.2. If*

   *(i)  stochastic FBF, i.e. $\Diamond_k = z_k$ and $\triangle_k = \eta_k$, with step size $\alpha_k \leq \alpha \leq 1/\sqrt{2}L$, or*

   *(ii)  stochastic FBFp, i.e. $\Diamond_k = w_{k-1}$ and $\triangle_k = \xi_{k-1}$, with step size $\alpha_k \leq \alpha \leq 1/3L$*

*is chosen, then for all $K \geq 1$ the averaged iterates $\bar{w}_K := \frac{\sum_{k=0}^{K-1} \alpha_k w_k}{\sum_{k=0}^{K-1} \alpha_k}$ fulfill*

$$\mathbb{E}[G_B(\bar{w}_K)] \leq \frac{D^2 + 24\sigma^2 \sum_{k=0}^{K-1} \alpha_k^2}{\sum_{k=0}^{K-1} \alpha_k},$$

*where $G_B$ is the restricted gap defined in (11).*

The above theorem exhibits a classical step size dependence (Robbins & Monro, 1951), yielding convergence for sequences $(\alpha_k)_{k \geq 0}$ that are square summable $\sum_{k=0}^\infty \alpha_k^2 < +\infty$ but not summable $\sum_{k=0}^\infty \alpha_k = +\infty$. Additionally, if in the setting of Theorem 3.2 the step size is chosen to be $\alpha_k = \alpha/\sqrt{k+1}$, a convergence rate can be obtained and is given by

$$\mathbb{E}[G_B(\bar{w}_K)] = \mathcal{O}\left(\frac{1}{\sqrt{K}}\right). \tag{12}$$

If the step size does not go to zero, the gap can usually not be expected to vanish either. However, we can still show decrease in the gap up to a residual stemming from the variance. In particular, for a constant step size $\alpha_k = \alpha$ we have

$$\mathbb{E}[G_B(\bar{w}_K)] \leq \frac{D^2}{\alpha K} + 24\sigma^2 \alpha. \tag{13}$$

Additionally, if the number of iterations $K$ is fixed beforehand, a conclusion similar to (12) can be obtained by choosing $\alpha = 1/\sqrt{K}$ in (13).

## 4 EXPERIMENTS

The aim of this section is to show how the use of methods with convergence guarantees, albeit only in the monotone setting, can yield better training performance for different architectures and objectives. In particular, we demonstrate that FBF can perform at least as good as EG although requiring less evaluations of the regularizers.

### 4.1 2D TOY EXAMPLE

Following Goodfellow (2016); Mescheder et al. (2018) and others we consider the canonical example $\min_x \max_y xy$, illustrating the cycling behavior of (even bilinear) minimax problems. We augment this approach by adding a nonsmooth L1-regularizer for one player, with $\kappa > 0$, resulting in

$$\min_{x \in \mathbb{R}} \max_{y \in [-1,1]} \kappa|x| + xy. \tag{14}$$

The aforementioned issue of GDA (and its proximal extension PGDA) cycling around the solution is highlighted in Figure 1. The other methods, for which we display the averaged iterates, however do converge to a solution and show a decrease in the restricted gap according to theory. Even though the proximal steps provide improvement towards the solution $(0, 0)$ and FBF only uses half the amount of evaluations compared to EG, it outperforms the competing algorithms.

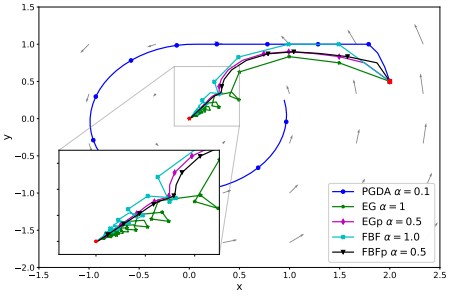
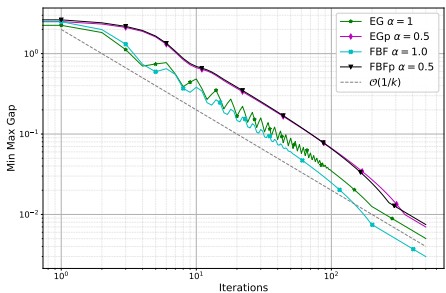

(a) Trajectories converging to solution.

(b) Restricted gap function.

Figure 1: A comparison of the methods presented in Section 2.3 applied to problem (14) with $\kappa = 0.01$. *PGDA* denotes (alternating) gradient descent ascent with proximal steps. As mentioned in the introduction it fails to converge. *EGp* denotes the method presented in Gidel et al. (2019) as extrapolation from the past. For the restricted gap we use $B_1 = B_2 = [-1, 1]$.

### 4.2 WGAN TRAINED ON CIFAR10

In this section we apply the above proposed techniques from monotone inclusions to the training of Wasserstein GANs employing DCGAN (Radford et al., 2015) and ResNet (He et al., 2016) architectures. All models are trained on the CIFAR10 dataset (Krizhevsky et al., 2009) which consists of 60,000 images in 10 different classes (with 50,000 training images and 10,000 test images) using an NVIDIA RTX 2080Ti GPU.

For the DCGAN experiments we work with the original WGAN formulation including weight clipping, since it includes regularizers innately (the indicator of a box for the weights of the discriminator). In addition we propose a modification of the WGAN formulation which replaces the box constraint on the discriminator's weights with an L1-regularization, under the name of *WGAN-L1*. This results in a *soft-thresholding* operation instead of the "harsh" clipping.

For the experiments on ResNet we use the WGAN-GP formulation (Gulrajani et al., 2017) which penalizes the norm of the gradient of the discriminator to enforce the Lipschitz constraint, together with spectral normalization of the weight matrices (Miyato et al., 2018) which can be seen as a projection.

Table 1: The best Inception Score (IS) and Fréchet Inception Distance (FID). The column denoted by *WGAN*, *WGAN-L1* and *WGAN-GP* refers to the standard formulation with weight clipping, our regularized implementation using the 1-norm and the formulation with gradient penalty and spectral normalization, respectively.

| Method | WGAN | | WGAN-L1 | | WGAN-GP | |
|---|---|---|---|---|---|---|
| | IS | FID | IS | FID | IS | FID |
| AltAdam1 | 4.12±.06 | 56.44±.62 | 4.43±.03 | 50.86±2.17 | 6.01±.31 | 28.11±3.65 |
| Extra Adam | 4.07±.05 | 56.67±.61 | 4.67±.11 | 47.24±1.21 | 6.58±.08 | 21.40±.58 |
| FBF Adam | 4.54±.04 | 45.85±.35 | 4.68±.16 | 46.60±.76 | 6.57±.10 | 21.22±1.29 |
| Opt. Adam | 4.35±.06 | 50.41±.46 | 4.63±.13 | 47.98±1.49 | 6.42±.10 | 23.01±.95 |

Given the ubiquity and dominance of Adam (Kingma & Ba, 2014) as an optimizer for many deep learning related training tasks, instead of using vanilla SGD we opt for Adam updates. This results in a method we call *FBF Adam*. Analogous approaches have been applied in Gidel et al. (2019) and Daskalakis et al. (2018) resulting in *Extra Adam* and *Optimistic Adam*, respectively. We compare the aforementioned methods with the status-quo in GAN training, namely alternating one Adam step for each network: *AltAdam1*.

Our hyperparameter search was limited to the step sizes when using the WGAN-L1 and WGAN-GP formulation, while all other parameters were kept the same as in Gidel et al. (2019); Boţ et al. (2020). It seems noteworthy that in the case of soft-thresholding bigger step sizes performed better with the only exception of AltAdam1.

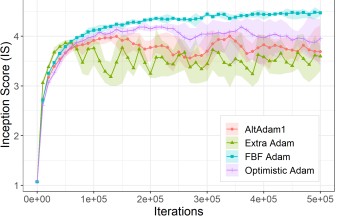 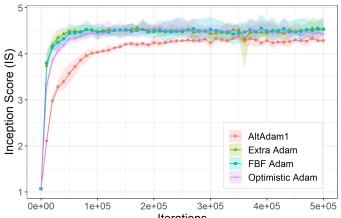

Figure 2: **Left:** Average and best/worst IS on the WGAN objective with weight clipping. **Right:** Average and best/worst IS on the WGAN-L1 objective using the proximal operator; The WGAN-L1 objective improves the IS in comparison to weight clipping and stabilizes the behavior of all considered methods during the training procedure.

The two evaluation metrics used are the Inception Score (IS, higher is better) (Salimans et al., 2016) and the Fréchet inception distance (FID, lower is better) (Heusel et al., 2017), both computed on 50,000 samples. In the case of the IS we use the updated and corrected implementation from Barratt & Sharma (2018). All results are averaged over 5 runs for each method.

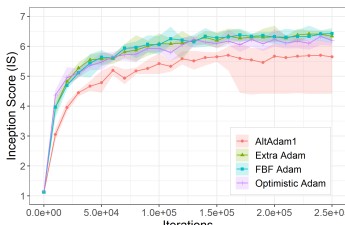 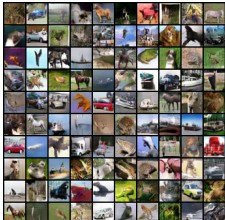 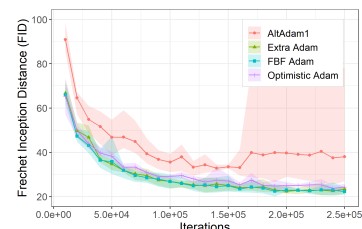

Figure 3: Average and best/worst results regarding IS (**left**) and FID (**right**) using ResNet architecture on the WGAN-GP objective including spectral normalization. **Middle:** Samples from the generator trained with FBF Adam.

In Table 1 the best IS and FID for each method are reported. FBF Adam performs at least as good as all considered competitors with respect to both evaluation metrics. One can also see that WGAN-L1

using the proximal operator improves the performance of all considered methods. Figure 2 shows the training progress regarding IS for each method and both problem formulations. The graphs suggest that making use of WGAN-L1 objective has a stabilizing effect during training, leading to a smoother and more consistent learning curve — a property that only FBF Adam seems to exhibit for weight clipping. Figure 3 as well as Table 1 show that for the WGAN-GP formulation FBF Adam maintains the improved performance of EG compared to GDA, while only requiring half the amount of spectral normalizations, resulting in time savings of up to 10% as reported in Miyato et al. (2018).

## 5 CONCLUSION

By highlighting the connection between GAN objectives and monotone inclusions, we are able to tackle their training via the Forward-Backward-Forward method which is known to converge to a solution for convex-concave minimax problems. We deepened this theoretical understanding by proving novel convergence rates in terms of the function values. We complement these rigorous considerations by promising practical results, indicating that application of FBF can lead to improved performance and saved computation time (compared to EG).

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

## A    DEFINITIONS

In Section 2.4 we require the regularizers to be proper, convex and lower semicontinuous which are common properties in convex analysis. We call a function $r : \mathbb{R}^m \to \mathbb{R} \cup \{+\infty\}$ *proper* if it is not constant $+\infty$, which means that it takes a finite value for at least a single point. In addition, we say that $r$ is *lower semicontinuous* if for all $z_0 \in \mathbb{R}^m$

$$\liminf_{z \to z_0} r(z) \geq r(z_0).$$

It is easy to see that if $C \subset \mathbb{R}^m$ is nonempty, closed and convex, then the indicator $\delta_C$ of this set, given by

$$\delta_C(z) = \begin{cases} 0 & \text{if } z \in C \\ +\infty & \text{otherwise} \end{cases}$$

fulfills the assumptions of being proper, convex and lower semicontinuous.

## B    ABOUT THE GAP FUNCTION

Typically in monotone inclusions, the distance to the set of solutions is used as a measure of quality of a given point due to the lack of more specific structure in general. Asymptotic convergence of the iterates has been established for FBF and FBFp in Bauschke & Combettes (2011, Proposition 27.13) and Malitsky & Tam (2020), respectively. Furthermore, no convergence rates can be expected without stronger monotonicity assumptions. We want to take into account the special structure of the monotone inclusion coming from the minimax problem (1). For this reason we use the following (restricted) *minimax gap*, common for saddle point problems, which for a point $(u, v)$ is given by

$$G_B(u, v) = \sup_{(x, y) \in B} \Psi(u, y) - \Psi(x, v). \tag{15}$$

For the general case, i.e. $F$ being an arbitrary monotone and Lipschitz operator this is connected to the other measure of optimality we use in (11), for $w \in \mathbb{R}^m$ given by

$$G_B(w) = \sup_{z \in B} \langle F(z), w - z \rangle + r(w) - r(z), \tag{16}$$

where we interpret the possible occurrence of $\infty - \infty$ as $+\infty$. It stems from the field of Variational Inequalities where such a function is also known as *merit function* (Nesterov, 2007). The relevance of the above two quantities will be made clear by the following statements.

**Theorem B.1.** *Let* $\Phi : \mathbb{R}^d \times \mathbb{R}^n \to \mathbb{R}$ *be continuously differentiable and* $f : \mathbb{R}^d \to \mathbb{R} \cup \{+\infty\}$, $h : \mathbb{R}^n \to \mathbb{R} \cup \{+\infty\}$ *be proper, convex and lower semicontinuous and* $B \subset \mathbb{R}^d \times \mathbb{R}^n$. *A point* $(x^*, y^*)$ *in the interior of* $B$ *solves the saddle point problem* (1) *if and only if its minimax gap* (15) *is zero,* $G_B(x^*, y^*) = 0$. *For all other elements of* $B$ *the gap is nonnegative.*

*Proof.* A saddle point $(x^*, y^*)$ clearly fulfills that $\sup_{(x,y) \in \mathbb{R}^d \times \mathbb{R}^n} \Psi(x^*, y) - \Psi(x, y^*) = 0$. On the other hand let $G_B(x^*, y^*) = 0$. For an arbitrary point $(x, y)$ we can choose $\alpha \in (0, 1)$ large enough such that $(u, v) := \alpha(x^*, y^*) + (1 - \alpha)(x, y)$ is in the interior of $B$. Therefore,

$$\Psi(x^*, v) - \Psi(u, y^*) = \Psi(x^*, \alpha y^* + (1 - \alpha)y) - \Psi(\alpha x^* + (1 - \alpha)x, y^*) \leq 0.$$

Using the convex-concave structure of $\Psi$ we deduce that

$$\alpha\Psi(x^*, y^*) + (1 - \alpha)\Psi(x^*, y) - \alpha\Psi(x^*, y^*) - (1 - \alpha)\Psi(x, y^*) \leq 0,$$

which implies that $\Psi(x^*, y) \leq \Psi(x, y^*)$. Since $(x, y)$ was chosen arbitrary $(x^*, y^*)$ is a saddle point. $\qquad \square$

Similarly, an analogous statement can be shown for (16). The proof, however is split up into multiple lemmas to highlight the connection to Variational Inequalities.

**Theorem B.2.** *Let $F : \mathbb{R}^m \to \mathbb{R}^m$ be monotone and continuous, $r : \mathbb{R}^m \to \mathbb{R} \cup \{+\infty\}$ proper, convex and lower semicontinuous and $B \subset \mathbb{R}^m$. A point $w^*$ in the interior of $B$ solves the monotone inclusion*

$$0 \in F(w) + \partial r(w) \tag{17}$$

*if and only if its restricted gap (16) is zero, $G_B(w^*) = 0$. For all other elements of $B$ the gap is nonnegative.*

Let the assumptions of Theorem B.2 hold true for the following lemmas as we break up the proof into separate statements. We do so by making use of the associated *Variational inequality (VI)*

$$\text{find } w \text{ such that} \quad \langle F(w), z - w \rangle + r(z) - r(w) \geq 0 \quad \forall z \in \mathbb{R}^m. \tag{18}$$

**Lemma B.3.** *The monotone inclusion (17) is equivalent to the VI (18).*

*Proof.* The equivalence of (17) and (18) follows immediately from the definition of the subdifferential of $r$. $\qquad \square$

The formulation (18) is typically referred to as the *strong* form of the VI, whereas

$$\text{find } w \text{ such that} \quad \langle F(z), z - w \rangle + r(z) - r(w) \geq 0 \quad \forall z \in \mathbb{R}^m, \tag{19}$$

is known as the *weak* formulation.

**Lemma B.4.** *Under the given assumptions the notion of weak and strong VI are equivalent.*

*Proof.* For the monotone operator $F$ it is clear that if $w^*$ is a solution to the strong formulation (18), it is also a solution to the weak formulation (19). In fact, if $F$ is continuous the reverse implication also holds true. To see this, let $w^*$ be a solution to the weak VI (19) and $z = \alpha w^* + (1 - \alpha)u$ for an arbitrary $u \in \mathbb{R}^m$ and $\alpha \in (0, 1)$, then

$$\langle F(\alpha w^* + (1 - \alpha)u), (1 - \alpha)(u - w^*) \rangle + r(\alpha w^* + (1 - \alpha)u) - r(w^*) \geq 0.$$

This implies by the convexity of $r$ that

$$(1 - \alpha)\langle F(\alpha w^* + (1 - \alpha)u), (u - w^*) \rangle + (1 - \alpha)(r(u) - r(w^*)) \geq 0.$$

By dividing by $(1 - \alpha)$ and then taking the limit $\alpha \to 1$ we obtain that $w^*$ is a solution of the strong form (18). $\qquad \square$

With the notion of VIs in mind, the above defined gap (16) becomes natural as it measures how much the statement of (19) is violated.

**Lemma B.5.** *$G_B$ is nonnegative on $B$ and zero for solutions of the weak VI.*

*Proof.* It is clear that $G_B(w) \geq 0$ for $w \in B$ as $z = w$ can be chosen in the supremum. On the other hand if $w^* \in B$ is a solution to the weak VI (19) then $G_B(w^*) = 0$. This follows from the fact that for a solution of (19) for all $z \in B$

$$\langle F(z), w^* - z \rangle + r(w^*) - r(z) \leq 0.$$

Therefore the supremum over the above expression in $z$ is also less than zero, but clearly zero is obtained for $z = w^*$. $\qquad \square$

For the reverse implication to hold true, we may not use points on the boundary of $B$.

**Lemma B.6.** *If a point $w^*$ in the interior of $B$ exhibits zero gap $G_B(w^*) = 0$, then it is a solution to the weak VI* (19).

*Proof.* Since $w^*$ is in the interior of $B$ we can, for an arbitrary $w \in \mathbb{R}^m$, choose $\alpha \in (0, 1)$ large enough such that $z := \alpha w^* + (1 - \alpha)w \in B$. Using this $z$ in the supremum of the gap we deduce that

$$\langle F(\alpha w^* + (1 - \alpha)w), w^* - \alpha w^* - (1 - \alpha)w \rangle + r(w^*) - r(\alpha w^* + (1 - \alpha)w) \leq 0.$$

This implies that

$$(1 - \alpha)\langle F(\alpha w^* + (1 - \alpha)w), w - w^* \rangle + (1 - \alpha)(r(w) - r(w^*)) \geq 0.$$

By dividing by $(1 - \alpha)$ and then taking the limit $\alpha \to 1$ we deduce that $w^*$ solves the strong form of the VI (18). $\qquad\square$

Now, we can turn to proving the theorem.

*Proof of Theorem B.2.* Combine Lemma B.3, B.4, B.5 and B.6. $\qquad\square$

## C    REFINED THEOREMS

Recall that restricted (unifying) gap function $G_B$ defined in (11) is computed with respect to a set $B \subset \mathbb{R}^m$ where $D := \sup_{w,z \in B} \|z - w\|$ denotes its diameter and it is assumed that $z_0 \in B$. Furthermore, the averaged iterates $\bar{w}_K$ for $K \geq 1$ are given by

$$\bar{w}_K := \frac{\sum_{k=0}^{K-1} \alpha_k w_k}{\sum_{k=0}^{K-1} \alpha_k}.$$

### C.1    DETERMINISTIC STATEMENTS

The convergence statement of Theorem 3.1 actually holds true not just for a constant step size as presented in Section 3, but for variable step sizes as well.

**Theorem C.1.** *Let $(w_k)_{k \geq 0}$ be the sequence generated by Algorithm 3.1. If*

*(i) FBF, i.e. $\Diamond_k = z_k$, with step size $0 < \alpha_k \leq \alpha \leq 1/L$, or*

*(ii) FBFp, i.e. $\Diamond_k = w_{k-1}$, with step size $0 < \alpha_k \leq \alpha \leq 1/2L$*

*is chosen, then for all $K \geq 1$*

$$G_B(\bar{w}_K) \leq \frac{D^2}{2 \sum_{k=0}^{K-1} \alpha_k}.$$

### C.2    STOCHASTIC STATEMENTS

We actually prove a slightly more general version of Theorem 3.2. In particular the step size can be chosen larger than initially claimed, however, at the cost of a worse constant.

**Theorem C.2.** *Let Assumption 1, 2 and 3 hold and let $(w_k)_{k \geq 0}$ be the sequence generated by FBF, i.e. Algorithm 3.2 with $\Diamond_k = z_k$ and $\triangle_k = \eta_k$. Let the step size $\alpha_k \leq \alpha < \frac{1}{L}$, then*

$$\mathbb{E}[G_B(\bar{w}_K)] \leq \frac{D^2 + 4(1 - \alpha^2 L^2)^{-1} \sigma^2 \sum_{k=0}^{K-1} \alpha_k^2}{2 \sum_{k=0}^{K-1} \alpha_k},$$

*for all $K \geq 1$.*

Theorem 3.2 (i) can be deduced from the above statement by using $\alpha = 1/\sqrt{2}L$ which yields that $(1 - \alpha^2 L^2)^{-1} = 2$.

**Theorem C.3.** *Let Assumption 1, 2 and 3 hold and let* $(w_k)_{k \geq 0}$ *be the sequence generated by FBFp, i.e. Algorithm 3.2 with* $\Diamond_k = w_{k-1}$ *and* $\triangle_k = \xi_{k-1}$*. Let the step size* $\alpha_k \leq \alpha < \frac{1}{2\sqrt{2}L}$*, then*

$$\mathbb{E}[G_B(\bar{w}_K)] \leq \frac{D^2 + 6\left(1 + \frac{4\alpha^2 L^2}{1 - 8\alpha^2 L^2}\right)\sigma^2 \sum_{k=0}^{K-1} \alpha_k^2}{\sum_{k=0}^{K-1} \alpha_k},$$

*for all* $K \geq 1$*.*

Theorem 3.2 (ii) is obtained from the above theorem by using the particular step size bound of $\alpha = 1/3L$, which yields that

$$\frac{4\alpha^2 L^2}{1 - 8\alpha^2 L^2} = 4.$$

Although, the step size in the refined statements Theorem C.2 and C.3 can be chosen arbitrarily close to $1/L$ and $1/(2\sqrt{2}L)$ for stochastic FBF and stochastic FBFp, respectively. This does not mean it should be — since the constant in the convergence rate deteriorates when the step size is close to its allowed upper bound.

## D  PROOFS

### D.1  PREPARATIONS

We introduce the notation connected to the strong formulation of the VI (18) associated to the monotone inclusion (9), given by

$$g(w, z) := \langle F(w), w - z \rangle + r(w) - r(z),$$

for $g : \mathbb{R}^m \times \mathbb{R}^m \to \mathbb{R} \cup \{+\infty\}$. Next we will establish the fact that this function can be used to bound the (restricted) unifying gap function, which we remind, is defined as

$$G_B(w) = \begin{cases} \sup_{(x,y) \in B} \Psi(u, y) - \Psi(x, v) & \text{if } F \text{ is (3)} \\ \sup_{z \in B} \langle F(z), w - z \rangle + r(w) - r(z) & \text{otherwise,} \end{cases}$$

where in the first case $(u, v) \in \mathbb{R}^d \times \mathbb{R}^n$ is identified with $w \in \mathbb{R}^m$. In particular the dimensions fulfill $d + n = m$, and $r(w)$ is given by $f(u) + h(v)$.

**Lemma D.1.** *It holds that for all* $K \geq 1$

$$\sup_{z \in B} \left\{ \frac{1}{\sum_{k=0}^{K-1} \alpha_k} \sum_{k=0}^{K-1} \alpha_k g(w_k, z) \right\} \geq G_B(\bar{w}_K).$$

*Proof.* First we will prove the case if $F$ is derived from a saddle point problem. Note that from the convex-concave structure of $\Phi$ we get that

$$\Phi(u, y) \leq \Phi(u, v) + \langle \nabla_y \Phi(u, v), y - v \rangle$$

and

$$\Phi(u, v) + \langle \nabla_x \Phi(u, v), x - u \rangle \leq \Phi(x, v).$$

By summing the two up we obtain

$$\Phi(u, y) - \Phi(x, v) \leq \left\langle \begin{array}{cc} -\nabla_x \Phi(u, v), & x - u \\ \nabla_y \Phi(u, v), & y - v \end{array} \right\rangle.$$

We can reformulate the above inequality in terms of $g$ to see that for $z = (x, y) \in \mathbb{R}^d \times \mathbb{R}^n$

$$\langle F(w), w - z \rangle \geq \Phi(u, y) - \Phi(x, v).$$

The statement of the first case is obtained by adding $r(w) - r(z)$ on both sides and using the fact that $\Psi$ is convex-concave.

If $F$ is a general monotone operator, then we use its monotonicity to deduce that

$$\langle F(w), w - z \rangle \geq \langle F(z), w - z \rangle.$$

The desired result follows from using the linearity of the inner product. □

**Notation.**    We denote the error of the stochastic estimator via

$$Z_k := F(\Diamond_k; \triangle_k) - F(\Diamond_k) \quad \text{and} \quad W_k := F(w_k; \xi_k) - F(w_k). \tag{20}$$

Furthermore, we will denote via $\mathbb{E}[\cdot \,|\, U]$, the conditional expectation with respect to the random variable $U$.

We will also need the following lemma.

**Lemma D.2.** *Let $(p_k)_{k \geq 0} \in \mathbb{R}^d$ be a given sequence and $(v_k)_{k \geq 0}$ recursively defined for all $k \geq 0$ by $v_{k+1} := v_k - p_k$ for some $v_0 \in \mathbb{R}^d$, then*

$$\sum_{k=0}^{K-1} \langle p_k, v_k - u \rangle \leq \frac{1}{2} \|v_0 - u\|^2 + \frac{1}{2} \sum_{k=0}^{K-1} \|p_k\|^2.$$

*Proof.* From the three point identity it follows immediately that

$$\langle p_k, v_k - u \rangle = \langle v_k - v_{k+1}, v_k - u \rangle = \frac{1}{2} \left( \|v_k - u\|^2 - \|v_{k+1} - u\|^2 + \|v_{k+1} - v_k\|^2 \right)$$

from which the statement of the lemma follows.    $\square$

## D.2    A UNIFIED DECREASE RESULT

We will start with a unifying proposition which covers the common parts of all convergence proofs.

**Proposition D.3.** *For a $\gamma > 0$ we have that for all $k \geq 0$ and $z \in \mathbb{R}^m$*

$$\alpha_k g(w_k, z) + \frac{1}{2} \|z_{k+1} - z\|^2 \leq \frac{1}{2} \|z_k - z\|^2 - \frac{1}{2} \|z_k - w_k\|^2 + \frac{1}{2} (1 + \gamma) \alpha_k^2 L^2 \|\Diamond_k - w_k\|^2$$
$$+ \alpha_k \langle W_k, z - w_k \rangle + (1 + \gamma^{-1}) \alpha_k^2 (\|W_k\|^2 + \|Z_k\|^2). \tag{21}$$

*Proof.* Let $k \geq 0$ and $z \in \mathbb{R}^m$ be arbitrary. Using the decomposition (20) it follows that

$$\langle \alpha_k F(w_k; \xi_k), w_k - z \rangle = \alpha_k \langle W_k, w_k - z \rangle + \alpha_k \langle F(w_k), w_k - z \rangle. \tag{22}$$

Since $\mathrm{prox}_{\alpha_k r} = (\mathrm{Id} + \alpha_k \partial r)^{-1}$ we deduce that

$$\langle z - w_k, w_k - z_k + \alpha_k F(\Diamond_k; \triangle_k) \rangle \geq \alpha_k (r(w_k) - r(z)). \tag{23}$$

Adding (22) and (23) gives that

$$\langle \alpha_k (F(w_k; \xi_k) - F(\Diamond_k; \triangle_k)) + z_k - w_k, w_k - z \rangle \geq \alpha_k \langle W_k, w_k - z \rangle + \alpha_k g(w_k, z),$$

which, using the definition of $z_{k+1}$, is equivalent to

$$\langle z - w_k, z_{k+1} - z_k \rangle \geq \alpha_k \langle W_k, w_k - z \rangle + \alpha_k g(w_k, z). \tag{24}$$

We estimate the inner product on the left side of the inequality by inserting and subtracting $z_k$ and using the three point identity twice to deduce

$$\langle z - w_k, z_{k+1} - z_k \rangle = \langle z - z_k + z_k - w_k, z_{k+1} - z_k \rangle$$
$$= \frac{1}{2} \left( \|z - z_k\|^2 - \|z_{k+1} - z\|^2 + \|z_{k+1} - w_k\|^2 - \|z_k - w_k\|^2 \right). \tag{25}$$

The first two summands are fine as they will telescope, so we are left with estimating $\|z_{k+1} - w_k\|^2$. By the definition of $z_{k+1}$ we have that

$$\|z_{k+1} - w_k\|^2 = \alpha_k^2 \|F(\Diamond_k; \triangle_k) - F(w_k; \xi_k)\|^2$$
$$= \alpha_k^2 \|F(\Diamond_k) - F(w_k) + Z_k - W_k\|^2$$
$$\leq (1 + \gamma) \alpha_k^2 \|F(\Diamond_k) - F(w_k)\|^2 + (1 + \gamma^{-1}) \alpha_k^2 \|Z_k - W_k\|^2 \tag{26}$$
$$\leq (1 + \gamma) \alpha_k^2 L^2 \|\Diamond_k - w_k\|^2 + 2(1 + \gamma^{-1}) \alpha_k^2 (\|Z_k\|^2 + \|W_k\|^2),$$

where we inserted and subtracted $F(\Diamond_k)$ and $F(w_k)$ and applied Young's inequality to deduce the result. Adding (26), (25) and (24) we conclude that

$$\alpha_k g(w_k, z) + \frac{1}{2} \|z_{k+1} - z\|^2 \leq \frac{1}{2} \|z_k - z\|^2 - \frac{1}{2} \|z_k - w_k\|^2 + \frac{1}{2} (1 + \gamma) \alpha_k^2 L^2 \|\Diamond_k - w_k\|^2$$
$$+ \alpha_k \langle W_k, z - w_k \rangle + (1 + \gamma^{-1}) \alpha_k^2 (\|W_k\|^2 + \|Z_k\|^2).$$
$$\square$$

### D.3 FORWARD-BACKWARD-FORWARD

*Proof for deterministic FBF, Theorem C.1 (i).* We start off by plugging $\diamondsuit_k = z_k$ into (21). Since $W_k = Z_k = 0$ we can use $\gamma \to 0$ to deduce that for all $k \geq 0$

$$\alpha_k g(w_k, z) + \frac{1}{2}\|z_{k+1} - z\|^2 \leq \frac{1}{2}\|z_k - z\|^2 - \frac{1}{2}(1 - \alpha_k^2 L^2)\|z_k - w_k\|^2.$$

From this it is clear that the step size is constrained by $\alpha \leq 1/L$ as stated in the theorem. By summing up from $k = 0$ to $K - 1$ and dividing by $\sum_{k=0}^{K-1} \alpha_k$ we obtain

$$\frac{1}{\sum_{k=0}^{K-1} \alpha_k} \sum_{k=0}^{K-1} \alpha_k g(w_k, z) \leq \frac{\|z_0 - z\|^2}{2\sum_{k=0}^{K-1} \alpha_k}.$$

The claimed statement is then derived by taking the supremum in $z$ over $B$ and applying Lemma D.1.
$\square$

*Proof for stochastic FBF, Theorem C.2.* Plugging $\diamondsuit_k = z_k$ and $\triangle_k = \eta_k$ into (21) gives for all $k \geq 0$

$$\alpha_k g(w_k, z) + \frac{1}{2}\|z_{k+1} - z\|^2$$
$$\leq \frac{1}{2}\|z_k - z\|^2 - \frac{1}{2}(1 - (1+\gamma)\alpha_k^2 L^2)\|z_k - w_k\|^2 + \alpha_k \langle W_k, z - v_k\rangle$$
$$+ \alpha_k \langle W_k, v_k - w_k\rangle + (1 + \gamma^{-1})\alpha_k^2(\|W_k\|^2 + \|Z_k\|^2).$$

By summing this inequality up and applying Lemma D.2 with $v_0 = z_0$, $p_k = -\alpha_k W_k$ and $v_{k+1} := v_k - p_k$ we deduce that

$$\sum_{k=0}^{K-1} \langle -\alpha_k W_k, v_k - z\rangle \leq \frac{1}{2}\|z_0 - z\|^2 + \frac{1}{2}\sum_{k=0}^{K-1} \alpha_k^2 \|W_k\|^2, \tag{27}$$

and therefore

$$\sum_{k=0}^{K-1} \alpha_k g(w_k, z) \leq \|z_0 - z\|^2 + \sum_{k=0}^{K} \alpha_k \langle W_k, v_k - w_k\rangle + 2(1 + \gamma^{-1})\alpha_k^2(\|W_k\|^2 + \|Z_k\|^2).$$

By choosing $\gamma$ such that $\alpha = (\sqrt{1+\gamma}L)^{-1}$ we deduce that $1 + \gamma^{-1} = 1/(1 - \alpha^2 L^2)$. Next, we take the supremum over $z \in B$ and the expectation to obtain

$$\mathbb{E}\left[\sup_{z \in B}\left\{\sum_{k=0}^{K-1} \alpha_k g(w_k, z)\right\}\right] \leq D^2 + 4(1 - \alpha^2 L^2)^{-1}\sigma^2 \sum_{k=0}^{K-1} \alpha_k^2,$$

where we used that

$$\mathbb{E}[\langle W_k, v_k - w_k\rangle] = \mathbb{E}\left[\mathbb{E}\left[\langle W_k, v_k - w_k\rangle \mid w_{[k]}, \xi_{[k-1]}\right]\right]$$
$$= \mathbb{E}\left[\left\langle \mathbb{E}\left[W_k \mid w_{[k]}, \xi_{[k-1]}\right], v_k - w_k\right\rangle\right] = 0,$$

with $\xi_{[k-1]} = (\xi_0, \ldots, \xi_{k-1})$ and $w_{[k]} = (w_0, \ldots, w_k)$. The final statement follows by dividing by $\sum_{k=0}^{K-1} \alpha_k$ and applying Lemma D.1.
$\square$

### D.4 FORWARD-BACKWARD-FORWARD-PAST

*Proof for deterministic FBFp, Theorem C.1 (ii).* We start off by plugging $\diamondsuit_k = z_k$ into (21). Since $W_k = Z_k = 0$ we can use $\gamma \to 0$ to conclude that for all $k \geq 0$

$$\alpha_k g(w_k, z) + \frac{1}{2}\|z_{k+1} - z\|^2 \leq \frac{1}{2}\|z_k - z\|^2 - \frac{1}{2}\|z_k - w_k\|^2 + \frac{1}{2}\alpha_k^2 L^2\|w_{k-1} - w_k\|^2. \tag{28}$$

Now we need to bound the term $\|w_{k-1} - w_k\|^2$ by $\|z_k - w_k\|^2$. Since

$$2\|z_k - w_k\|^2 + 2\|z_k - w_{k-1}\|^2 \geq \|w_k - w_{k-1}\|^2 \tag{29}$$

we have for all $k \geq 1$

$$
\|z_k - w_k\|^2 \geq -\|z_k - w_{k-1}\|^2 + \frac{1}{2}\|w_{k-1} - w_k\|^2
$$
$$
\geq -\alpha_{k-1}^2 L^2 \|w_{k-1} - w_{k-2}\|^2 + \frac{1}{2}\|w_{k-1} - w_k\|^2
$$

(30)

whereas for $k = 0$, since $w_{-1} = z_0$, we have that

$$
\|z_0 - w_0\|^2 = \|w_{-1} - w_0\|^2.
$$

(31)

Plugging (31) into (28) for $k = 0$ we get that

$$
\alpha_0 g(w_0, z) + \frac{1}{2}\|z_1 - z\|^2 + \frac{1}{2}(1 - \alpha_0^2 L^2)\|w_0 - w_{-1}\|^2 \leq \frac{1}{2}\|z_0 - z\|^2.
$$

(32)

Plugging (30) into (28) we get that for all $k \geq 1$

$$
\alpha_k g(w_k, z) + \frac{1}{2}\|z_{k+1} - z\|^2 + \frac{1}{2}\left(\frac{1}{2} - \alpha_k^2 L^2\right)\|w_k - w_{k-1}\|^2
$$
$$
\leq \frac{1}{2}\|z_k - z\|^2 + \frac{1}{2}\alpha_{k-1}^2 L^2 \|w_{k-1} - w_{k-2}\|^2.
$$

(33)

In order to be able to telescope we need to ensure that for all $k \geq 0$

$$
\left(\frac{1}{2} - \alpha_k^2 L^2\right) \geq \alpha_k^2 L^2.
$$

This is equivalent to the condition $\alpha_k \leq 1/2L$ which was required in the statement of the theorem. Now we sum up (33) from $k = 1$ to $K - 1$ which yields

$$
\sum_{k=1}^{K-1} \alpha_k g(w_k, z) + \frac{1}{2}\|z_K - z\|^2 + \frac{1}{2}\left(\frac{1}{2} - \alpha_{K-1}^2 L^2\right)\|w_{K-1} - w_{K-2}\|^2
$$
$$
\leq \frac{1}{2}\|z_1 - z\|^2 + \frac{1}{2}\alpha_0^2 L^2 \|w_0 - w_{-1}\|^2.
$$

(34)

Adding (34) and (32) and dividing by $\sum_{k=0}^{K-1} \alpha_k$ to deduce

$$
\frac{1}{\sum_{k=0}^{K-1} \alpha_k} \sum_{k=0}^{K-1} \alpha_k g(w_k, z) \leq \frac{\|z_0 - z\|^2}{2\sum_{k=0}^{K-1} \alpha_k},
$$

where we used that $1 - \alpha_0^2 L^2 \geq \alpha_0^2 L^2$ to get rid of $\|w_0 - w_{-1}\|^2$. The final statement follows by taking the supremum in $z$ over $B$ and applying Lemma D.1. $\qquad\square$

*Proof for stochastic FBFp, Theorem C.3.* By using $\Diamond_k = w_{k-1}$ we deduce from (21) for all $k \geq 0$ that

$$
\alpha_k g(w_k, z) + \frac{1}{2}\|z_{k+1} - z\|^2 \leq \frac{1}{2}\|z_k - z\|^2 - \frac{1}{2}\|z_k - w_k\|^2 + \frac{1}{2}(1 + \gamma)\alpha_k^2 L^2 \|w_{k-1} - w_k\|^2
$$
$$
+ \alpha_k \langle W_k, z - w_k \rangle + 2(1 + \gamma^{-1})\alpha_k^2(\|W_k\|^2 + \|Z_k\|^2).
$$

As in (27) we can split $\langle \alpha_k W_k, z - w_k \rangle$ into $\langle \alpha_k W_k, z - v_k \rangle + \langle \alpha_k W_k, v_k - w_k \rangle$ and use Lemma D.2 to deduce

$$
\sum_{k=0}^{K-1} \alpha_k g(w_k, z) \leq \|z_0 - z\|^2 - \sum_{k=0}^{K-1} \left(\frac{1}{2}\|z_k - w_k\|^2 + \frac{1}{2}(1 + \gamma)\alpha_k^2 L^2 \|w_{k-1} - w_k\|^2 \right.
$$
$$
\left. + \langle \alpha_k W_k, v_k - w_k \rangle + 3(1 + \gamma^{-1})\alpha_k^2(\|W_k\|^2 + \|Z_k\|^2)\right).
$$

Taking now the supremum over $z \in B$ and then the expectation we conclude that the inequality

$$
\mathbb{E}\left[\sup_{z \in B}\left\{\sum_{k=0}^{K-1} \alpha_k g(w_k, z)\right\}\right] \leq D^2 - \frac{1}{2}\sum_{k=0}^{K-1}\left(\|z_k - w_k\|^2 - (1 + \gamma)\alpha_k^2 L^2 \|w_{k-1} - w_k\|^2\right)
$$
$$
+ 3(1 + \gamma^{-1})\sigma^2 \sum_{k=0}^{K-1} \alpha_k^2
$$

(35)

holds. Let from now on $k \geq 1$ as we will treat the case $k = 0$ separately. Using (29) we deduce that

$$
\begin{aligned}
\|z_k - w_k\|^2 &\geq -\|z_k - w_{k-1}\|^2 + \frac{1}{2}\|w_{k-1} - w_k\|^2 \\
&\geq -\alpha_{k-1}^2 \|F(w_{k-1}; \xi_{k-1}) - F(w_{k-2}; \xi_{k-2})\|^2 + \frac{1}{2}\|w_{k-1} - w_k\|^2.
\end{aligned}
\tag{36}
$$

Now we bound the difference of the two estimators by inserting $\pm F(w_{k-1}), \pm F(w_{k-2})$ and applying the inequality $\|a + b + c\|^2 \leq 3(\|a\|^2 + \|b\|^2 + \|c\|^2)$ which yields

$$
\|F(w_{k-1}; \xi_{k-1}) - F(w_{k-2}; \xi_{k-2})\|^2 \leq 3\|W_{k-1}\|^2 + 3\|W_{k-2}\|^2 + 3\|F(w_{k-2}) - F(w_{k-1})\|^2.
$$

We conclude that

$$
\mathbb{E}\left[\|F(w_{k-1}; \xi_{k-1}) - F(w_{k-2}; \xi_{k-2})\|^2\right] \leq 6\sigma^2 + 3L^2 \mathbb{E}\|w_{k-1} - w_{k-2}\|^2.
\tag{37}
$$

Using (37) in (36) we deduce that

$$
\mathbb{E}\|z_k - w_k\|^2 \geq -\alpha_{k-1}^2(6\sigma^2 + 3L^2 \mathbb{E}\|w_{k-1} - w_{k-2}\|^2) + \frac{1}{2}\mathbb{E}\|w_{k-1} - w_k\|^2,
\tag{38}
$$

whereas for $k = 0$ we have (31). Now we plug (38) into (35) to conclude that

$$
\begin{aligned}
\mathbb{E}&\left[\sup_{z \in B}\left\{\sum_{k=0}^{K-1} \alpha_k g(w_k, z)\right\}\right] \\
&\leq D^2 - \frac{1}{2}\sum_{k=1}^{K-1}\left(-3\alpha_{k-1}^2 L^2 \mathbb{E}\|w_{k-1} - w_{k-2}\|^2 + \left(\frac{1}{2} - (1+\gamma)\alpha_k^2 L^2\right)\|w_{k-1} - w_k\|^2\right) \\
&\quad + \frac{1}{2}((1+\gamma)\alpha_0^2 L^2 - 1)\|w_{-1} - w_0\|^2 + 6(1+\gamma^{-1})\sigma^2 \sum_{k=0}^{K-1} \alpha_k^2
\end{aligned}
\tag{39}
$$

From this we conclude that in order to be able to telescope we need to enforce

$$
\left(\frac{1}{2} - (1+\gamma)\alpha_k^2 L^2\right) \geq 3\alpha_k^2 L^2
$$

which is equivalent to

$$
\frac{1}{2(4+\gamma)} \geq \alpha_k^2 L^2.
$$

Since $\alpha_k \leq \alpha$, we can ensure this by choosing $\gamma$ such that

$$
\frac{1}{2(4+\gamma)} = \alpha^2 L^2.
\tag{40}
$$

With (40) in place conclude from (39) that the inequality

$$
\begin{aligned}
\mathbb{E}&\left[\sup_{z \in B}\left\{\sum_{k=0}^{K-1} \alpha_k g(w_k, z)\right\}\right] \\
&\leq D^2 + \frac{1}{2}((4+\gamma)\alpha_0^2 L^2 - 1)\|w_{-1} - w_0\|^2 + 6(1+\gamma^{-1})\sigma^2 \sum_{k=0}^{K-1} \alpha_k^2
\end{aligned}
$$

Using the fact that $3\alpha_0^2 L^2 \leq 1 - (1+\gamma)\alpha_0^2 L^2$ from (40) to discard the $\|w_0 - w_{-1}\|^2$ term, yields

$$
\mathbb{E}\left[\sup_{z \in B}\left\{\sum_{k=0}^{K-1} \alpha_k g(w_k, z)\right\}\right] \leq D^2 + 6(1+\gamma^{-1})\sigma^2 \sum_{k=0}^{K-1} \alpha_k^2
\tag{41}
$$

Through (40), we can estimate

$$
\frac{1}{\gamma} = \frac{2\alpha^2 L^2}{1 - 8\alpha^2 L^2}.
\tag{42}
$$

Plugging (42) into (41), dividing by $\sum_{k=0}^{K-1} \alpha_k$ and applying Lemma D.1, deduces the final statement. $\qquad\square$

# E ARCHITECTURE

## E.1 DCGAN

Table 2: DCGAN architecture used for the CIFAR10 experiments.

| **Generator** |
|---|
| *Input:* $z \in \mathbb{R}^{128} \sim \mathcal{N}(0, I)$ |
| Linear $128 \rightarrow 512 \times 4 \times 4$ |
| Batch Normalization |
| ReLU |
| transposed conv. (kernel: $4 \times 4$, $512 \rightarrow 256$, stride: 2, pad: 1) |
| Batch Normalization |
| ReLU |
| transposed conv. (kernel: $4 \times 4$, $256 \rightarrow 128$, stride: 2, pad: 1) |
| Batch Normalization |
| ReLU |
| transposed conv. (kernel: $4 \times 4$, $128 \rightarrow 3$, stride: 2, pad: 1) |
| $Tanh(\cdot)$ |

| **Discriminator** |
|---|
| *Input:* $x \in \mathbb{R}^{3 \times 32 \times 32}$ |
| conv. (kernel: $4 \times 4$, $1 \rightarrow 64$, stride: 2, pad: 1) |
| LeakyReLU (negative slope: 0.2) |
| conv. (kernel: $4 \times 4$, $64 \rightarrow 128$, stride: 2, pad: 1) |
| Batch Normalization |
| LeakyReLU (negative slope: 0.2) |
| conv. (kernel: $4 \times 4$, $128 \rightarrow 256$, stride: 2, pad: 1) |
| Batch Normalization |
| LeakyReLU (negative slope: 0.2) |
| Linear $128 \times 4 \times 4 \times 4 \rightarrow 1$ |

## E.2 RESNET

Table 3: ResNet architecture used for the CIFAR10 experiments.

| **Generator** |
|---|
| *Input:* $z \in \mathbb{R}^{128} \sim \mathcal{N}(0, I)$ |
| Linear $128 \rightarrow 128 \times 4 \times 4$ |
| ResBlock $128 \rightarrow 128$ |
| ResBlock $128 \rightarrow 128$ |
| ResBlock $128 \rightarrow 128$ |
| ReLU |
| transposed conv. (kernel: $3\times3$, $128 \rightarrow 3$, stride: 1, pad: 1) |
| $Tanh(\cdot)$ |

| **Discriminator** |
|---|
| *Input:* $x \in \mathbb{R}^{3 \times 32 \times 32}$ |
| ResBlock $3 \rightarrow 128$ |
| ResBlock $128 \rightarrow 128$ |
| ResBlock $128 \rightarrow 128$ |
| ResBlock $128 \rightarrow 128$ |
| Linear $128 \rightarrow 1$ |

## F    HYPERPARAMETERS

For the WGAN formulation with *weight clipping*, see Table 4, we used the extensively tuned hyperparameters from Gidel et al. (2019) for ExtraAdam, Adam1 and OptimisticAdam. Note that our values of the Inception Score (IS) differ from the ones reported in Gidel et al. (2019) as we use the newer implementation of the IS proposed in Barratt & Sharma (2018). For FBF-Adam we tuned the step size and kept all other hyperparameters equal.

Table 4: Hyperparameters used for the WGAN formulation (with weight clipping).

| **(DCGAN) WGAN Hyperparameters** | |
| --- | --- |
| Batch size | $= 64$ |
| Number of generator updates | $= 500,000$ |
| Adam $\beta_1$ | $= 0.5$ |
| Adam $\beta_2$ | $= 0.9$ |
| Weight clipping for the discriminator | $= 0.01$ |
| Learning rate for discriminator | $= 5 \times 10^{-4}$ (Extra Adam) |
| | $= 2 \times 10^{-4}$ (AltAdam1, FBF Adam, Optim. Adam) |
| Learning rate for generator | $= 5 \times 10^{-5}$ (Extra Adam) |
| | $= 2 \times 10^{-5}$ (AltAdam1, FBF Adam, Optim. Adam) |

For our newly proposed WGAN-L1 formulation using 1-Norm regularization, see Table 5, we limited the hyperparameter search to the step sizes, with the values in Table 4 as initial guesses. We chose the value performing the best in terms of IS and FID for a sample seed. All other parameters were kept the same as in Gidel et al. (2019); Boţ et al. (2020).

Table 5: Hyperparameters used for the WGAN-L1 formulation (with soft thresholding).

| **(DCGAN) WGAN-L1 Hyperparameters** | |
| --- | --- |
| Batch size | $= 64$ |
| Number of generator updates | $= 500,000$ |
| Adam $\beta_1$ | $= 0.5$ |
| Adam $\beta_2$ | $= 0.9$ |
| L1 regularization for the discriminator | $= 1 \times 10^{-4}$ |
| Learning rate for discriminator | $= 1 \times 10^{-3}$ (FBF Adam, Extra Adam) |
| | $= 5 \times 10^{-4}$ (Optim. Adam) |
| | $= 2 \times 10^{-4}$ (AltAdam1) |
| Learning rate for generator | $= 1 \times 10^{-4}$ (FBF Adam, Extra Adam) |
| | $= 5 \times 10^{-5}$ (Optim. Adam) |
| | $= 2 \times 10^{-5}$ (AltAdam1) |

For the experiments based on the WGAN-GP formulation including spectral normalization we limited the hyperparameter search to the step sizes, with the values recommended in Gidel et al. (2019) as initial guesses. We used a single power iteration for the spectral normalization as suggested in Miyato et al. (2018) and reduced the number of generator updates by a factor of two to ease the computational burden.

Table 6: Hyperparameters used for the WGAN-GP formulation (with spectral normalization).

| (ResNet) WGAN-GP Hyperparameters | |
|---|---|
| Batch size | $= 64$ |
| Number of generator updates | $= 250,000$ |
| Adam $\beta_1$ | $= 0.5$ |
| Adam $\beta_2$ | $= 0.9$ |
| Gradient penalty | $= 10$ |
| Power iterations for spectral normalization | $= 1$ |
| Learning rate for discriminator | $= 5 \times 10^{-4}$ (FBF Adam, Extra Adam, Optim. Adam) $= 2 \times 10^{-5}$ (AltAdam1) |
| Learning rate for generator | $= 5 \times 10^{-4}$ (FBF Adam, Extra Adam, Optim. Adam) $= 2 \times 10^{-5}$ (AltAdam1) |

