# OpenReview forum: "Two steps at a time --- taking GAN training in stride with Tseng's method"
_ICLR.cc/2021/Conference — Reject_

### Official Review · AnonReviewer4 · 2020-10-13
**Unifying Optimistic Gradient Descent Ascent and Forward Backward Forward For Convex-Concave Optimization**

**Rating:** 6
**Confidence:** 3

**Review:**

The authors in this paper, inspired by the applications of min-max optimization in GANs, study the problem of min-max optimization for convex-concave functions. The main contribution of the paper is proving novel convergence results for Forward-Backward-Forward (FBF) algorithms as well as Optimistic Gradient Descent Ascent (OGDA) based on tools from monotone inclusion problems. Their convergence results cover both deterministic and stochastic settings and the rates of convergence for suitably chosen gap function are non-asymptotic. Finally, they apply their algorithms both on toy problems but also on training GANs on CIFAR-10.

Pros:
1) To the best of my knowledge, the connection between OGDA and monotone inclusion problems is new.
2) Convergence results are non asymptotic for the specified gap function.

Cons:
1) The connection of this work with GANs is a bit tenuous because, as the authors also acknowledge, training GANs is a non-convex non-concave min-max problem. The authors should try to express why they believe their  unification result informs practical applications.
2) This lack of connection is reflected in the experimental section as well. Most experiments re-establish that optimism, extragradient updates or regularization are beneficial for min-max optimization, observations that are already widely known. Again, any experimental insight that is particular to this work, would go a long way towards closing this gap.
3) It is not clear that the connection of OGDA to FBF and monotone inclusion provides any new insights about the convergence properties of either method.  It would be very helpful, if the authors provided any additional intuition why their result could be used to answer open questions related to OGDA or FBF.  For example, last iterate convergence of OGDA is still an open problem even in convex-concave problems.
4) While the gap function used allows the authors to provide non asymptotic guarantees, the intuition behind this gap function when its value is non-zero is unclear. Does this gap function have any game theoretic interpretation?

For now, I am assigning a weak reject score mainly because it is unclear to me if there are significant implications of this unification result either in theory or practice. I am willing to increase my score substantially if the authors provide additional details that address my concerns outlined above.

---------------------------------
Post-Rebuttal evaluation.

I would like to thank the authors for their detailed answers, especially regarding the interpretation of the gap function.
Based on their answers, I decided to increase my score to a 6.

---

> ### Author Response · Authors · 2020-11-13
> **Response to Reviewer4**
>
> Dear reviewer, thank you for your time and remarks. For your convenience we highlighted all the changes in the pdf in red.
>
> Since some of the points raised in Cons. 1-3 are connected we try and respond to them at the same time.
>
> > The authors should try to express why they believe their unification result informs practical applications.
> > It is not clear that the connection of OGDA to FBF and monotone inclusion provides any new insights about the convergence properties of either method.
> > [...] any experimental insight that is particular to this work
>
> The unification result itself may not provide much practical benefit although it sheds some light on the stepsize requirements of the different methods (see the fourth comment to R3). The benefit of our work for practical applications lies more in the treatment of regularizers and the fact that FBF requires less evaluations (see the first reply to R3) and is thus able to save computation time (compared to EG) without sacrificing performance.
>
> > last iterate convergence of OGDA is still an open problem even in convex-concave problems
>
> As far as we know asymptotic convergence of the iterates is known for OGDA.
> Regarding convergence *rates* for the (last) iterate(s) we don't think it is possible to obtain such a statement without the use of more restrictive assumptions (see last comment to R3). If talking about rates for the gap function but in terms of the last iterate (instead of the ergodic), this constitutes indeed an interesting and possibly obtainable (albeit more difficult) statement, but was outside the scope of our work.
>
> > the intuition behind this gap function [...] is unclear
>
> Thank you this very valid comment. We added a clarifying remark and intuition in the updated version of our manuscript. Please see also the reply to R1 were we give a short summery.

---

### Official Review · AnonReviewer2 · 2020-10-19
**Report on "Two steps at a time --- taking GAN training in stride with Tseng's method"**

**Rating:** 4
**Confidence:** 5

**Review:**

Summary: This paper introduces the forward-backward-forward splitting for variational inequalities. The main results are an asymptotic convergence results and a non-asymptotic convergence results using a restricted merit function. A new method, FBFp, is introduced and studied. Complete proofs are given. Preliminary numerical results obtained by training GANs are reported.

Pros:
+ complete proofs
+ A new stochastic operator splitting method based on Tseng's FBF is introduced in which the operator needs to be evaluated once per iteration. This splitting, called FBFp, is indeed new, and has the potentially of being of practical relevance.
+ Preliminary numerical results on standard GAN architectures.

Cons:
- The paper takes a long time until it becomes clear what actually the monotone inclusion looks like. It seems that the problem of interest is formulation in eq. (9), preceded by a long and unnecessary discussion about existing solvers. It would have been much more accurate to simply start with the problem formulation, then propose your solution method, followed by a critical explanation of the contribution.
- p.3 claim that FBF has not been rigorously analyzed for saddle point problems. This is of course not true. Even the original paper by Tseng (A MODIFIED FORWARD-BACKWARD SPLITTING METHOD FOR MAXIMAL MONOTONE MAPPINGS, SICON 2000) discusses the application to saddle point problems. See Example 5 in that paper.
- The stochastic FBF has been studied in Bot et al. Mini-batch Forward-Backward-Forward Methods for solving Stochastic Variational inequalities, forthcoming in Stochastic Systems. Note that the Arxive version of that paper is available since 2019. Overall, the paper contains only marginal contributions to the state-of-the-art.
- Only convergence rates for the ergodic average is provided. It is known that the ergodic average might destroy important features of the true solution, such as sparsity. For SFBF we know non-asymptotic convergence rates of the last iterate. This is not mentioned at all.
- I have some doubts that the restricted merit function is the appropriate one here. Note if the aim is to solve the VI over an unconstrained domain, then FBF coincides with EG, and there is nothing to be analyzed. The interesting case is thus only the constrained case. These constraints are usually encoded in the non-smooth part of of eq. (8), so there is no need to write this explicitly. In my opinion it would therefore be cleaner to assume at the outset that the domain of $F+\partial r$ is bounded. The gap function used can in fact be traced back to Facchinei & Pang (2003) and is most likely even longer in use than that.

---

> ### Author Response · Authors · 2020-11-13
> **Response to Reviewer2**
>
> Dear reviewer, thank you for your time and interest. For your convenience we highlighted all the changes in the pdf in red.
>
> > The paper takes a long time until it becomes clear what actually the monotone inclusion looks like.
>
> Since monotone inclusions are not a well known object in the ML community we wanted to make sure that the concept is properly motivated.
>
> > preceded by a long and unnecessary discussion about existing solvers
>
> Due to the similarities of the existing methods we think that this exposition is necessary in order to not create confusion. First of all it seems very natural to us to compare FBF with EG since the two methods are so closely related. Secondly, by recycling previous gradients we provide a novel intuition for OGDA/FRB, which we think is relevant and requires us to mention said methods.
>
> > p.3 claim that FBF has not been rigorously analyzed for saddle point problems. This is of course not true.
>
> We meant, but did not write explicitly, this statement to be in terms of function values / gap. We added this clarifying specification in the pdf on page 3.
> At the same time we would like to point out that in the contribution paragraph and in the conclusion this was already stated more precisely before.
>
> > The stochastic FBF has been studied in [...]
>
> We added the reference and clarified the differences with the mentioned paper. In particular, the cited article does not deal with the minimax setting specifically and is therefore not able to make statements about the gap function.
>
> > For SFBF we know non-asymptotic convergence rates of the last iterate. This is not mentioned at all.
>
> Thank you for pointing this out. We mentioned this in the updated version of our preprint and clarified that these rates are in terms of fixed point residual, which is a rather general notion and does not exploit the special structure of the problem. Another key difference between our work and the mentioned paper is that we do not rely on minibatch sizes that go to infinity.
>
> > if the aim is to solve the VI over an unconstrained domain, then FBF coincides with EG [...]
> In my opinion it would therefore be cleaner to assume at the outset that the domain of $F+\partial r$ is bounded
>
> We do not see a convincing reason to assume that the domain of the problem is bounded. In fact there are many interesting regularizers which do not have a bounded domain. Note further that EG and FBF are not the same method in this regularized (but unconstrained) setting.

---

### Official Review · AnonReviewer3 · 2020-10-28
**Contributions seem incremental**

**Rating:** 4
**Confidence:** 4

**Review:**

Summary: This work studies minimax optimization (a.k.a saddle-point problems) with nonsmooth regularizers. By leveraging the monotone operator theory, the authors propose to use the forward-backward-forward method so as to avoids the notorious limit cycling problem. The classical FBF method requires two gradient evaluations per step, the authors introduce a new algorithm which reuses the past gradient in the same way as OGDA. In the setting of convex-concave minimax optimization, the authors claim to prove novel convergence rates for both methods.

Review: This paper is well-written, though I think the difference with extra-gradient can be made much clearer. For example, the authors could compare FBF with EG for a toy example like Eqn. (13). As far as I can tell, the main difference with extra-gradient is just the regularization term. From this standpoint, the authors should clarify the motivations and importance of adding regularizers. In the current version of the paper, section 2.4 is vague and doesn't explain the role of regularizers well in GAN training.
Besides, a similar convergence analysis has been done for EG and OGDA (the authors don't even cite properly), so I believe the convergence analysis of this paper is not novel. Given these reasons, I suggest the rejection of this paper and give a score of 4.

Comments:
- It was shown in [1] that the ergodic (averaged) iterates of extra-gradient converge with a rate of O(1/k) which is the same as FBF. Could the authors clarify the novelty/difference of your proof technique for FBF?

- It was shown in [2] that extra-gradient is NOT robust to gradient noise in convex-concave minimax optimization. Could the authors comment on that and explain why FBF could achieve the rate of O(1/sqrt(k)) in the stochastic setting (while EG fails)?

- In GAN training, we typically care more about the last iterate since averaging could actually hurt the performance when the loss surface is highly nonconvex. Is it possible to derive the last iterate convergence rate?


I'm willing to increase my rating if the authors could resolve some of my concerns, especially my concern on the novelty of the analysis.


-------------
**I've read the authors' response. I'm still concerned with the novelty of the paper given there are similar results for EG/OGDA. Therefore, I stick to my original rating.**

References:

[1] Convergence Rate of O(1/k) for Optimistic Gradient and Extra-gradient Methods in Smooth Convex-Concave Saddle Point Problems, 2019.

[2] Explore Aggressively, Update Conservatively: Stochastic Extragradient Methods with Variable Stepsize Scaling, 2020.

---

> ### Author Response · Authors · 2020-11-13
> **Response to Reviewer3**
>
> Dear reviewer, thank you for your interest and remarks. For your convenience we highlighted all the changes in the pdf in red.
>
> > the authors could compare FBF with EG for a toy example like Eqn. (13)
>
> The two methods are compared for this problem in Figure 1. The difference between the two methods is not so much the regularization term (which is part of the objective function) but how it is treated: per iteration EG requires two evaluations of the projection (prox) corresponding to the regularizer, while FBF only needs one.
> Note that in the absence of regularizers or constraints (in the problem formulation) EG and FBF reduce to the same method.
>
> > the authors should clarify the motivations and importance of adding regularizers. In the current version of the paper, section 2.4 is vague and doesn't explain the role of regularizers well in GAN training.
>
> We added some remarks about regularizers used for GAN training in Section 2.4 mentioning the box constraints in the original WGAN formulation and spectral normalization for SN-GAN.
>
> > It was shown in [1] that the ergodic (averaged) iterates of extra-gradient converge with a rate of O(1/k)
>
> Thank you for pointing this article out, which we were not aware of. We added the reference and adapted the text accordingly. Note however that our result still provides a generalization in the sense that we also cover regularizers which encompass, but are not limited to constraints.
>
> > a similar convergence analysis has been done for EG and OGDA.
> > Could the authors clarify the novelty/difference of your proof technique for FBF?
>
> Since OGDA, EG and FBF are similar they naturally rely on similar proof techniques. Through our unified approach it becomes easier to draw a distinction between and highlight the similarities of OGDA and FBF. Because of this we are able to pinpoint where the different stepsize requirements arise (OGDA only allows for half as big of a stepsize compared to FBF, and thus possibly negate every saving stemming from the reduced amount of gradient evaluations..). In particular, it indicates why OGDA tends to requires a smaller stepsize than FBF/EG in applications.
>
> > It was shown in [2] that extra-gradient is NOT robust to gradient noise in convex-concave minimax optimization.
> Could the authors comment on that and explain why FBF could achieve the rate of $O(1/sqrt(k))$ in the stochastic
> setting (while EG fails)?
>
> Reference [2] itself mentions that EG exhibits the same $O(1/\sqrt{k})$ rate, see Table 1 in that paper. These two seemingly conflicting statements stem in our opinion from the fact that one is with respect to the sequence of (last) iterates itself and the other one is in terms of the ergodic (averaged) ones. The mentioned issue is of course relevant and a version of FBF which is robust to noise would be interesting.
>
> > Is it possible to derive the last iterate convergence rate?
>
> This is unknown and even unlikely as convergence rates for the iterates can typically only be obtained under more restrictive assumptions as strong convexity/monotonicity (or error bounds).

---

> > ### Comment · AnonReviewer3 · 2020-11-13
> > **Last iterate convergence rates**
> >
> > Please see the following paper for the last iterate analysis:
> >
> > - Last Iterate is Slower than Averaged Iterate in Smooth Convex-Concave Saddle Point Problems

---

> > > ### Author Response · Authors · 2020-11-14
> > > **we meant distance to solution**
> > >
> > > Thank you for pointing out the article. We thought 'distance of the last iterate to the solution set' was meant, while the mentioned paper uses 'norm of the gradient' and 'gap'. Unfortunately we do not know whether this analysis can be adapted to the regularized setting, even under the stronger assumption of Lipschitz second derivatives.

---

### Official Review · AnonReviewer1 · 2020-10-28
**Many unclear or doubting points**

**Rating:** 4
**Confidence:** 3

**Review:**

In this paper,
the authors first formulate the optimization problem of GANs as an abstract minimax problem (Equation(1)).
As compared to the original optimization objective of Goodfellow's GANs,
there is an additional term $h(y)$.
Why do you introduce $h(y)$ here? Just for facilitating the adoption of the MONOTONE INCLUSIONS on it?
The authors should provide a clear explanation about this point.

Later the author restrict Equation(1) to a deterministic version,
which means that the noise input of GANs will no longer be considered.
The noise input is an important ingredient of GANs.
Despite many new variants of the GANs,
at least, the noise input is import for the Goodfellow's GANs, which is adopted in this submission as a staring point.
It is very hard for me to decide whether this simplification is appropriate.
Except Algorithm 3.2 and Theorem 3.2 which suddenly provide stochastic versions,
all the following results are on this deterministic version.
In my opinion, the authors should provide more explanations on this point.

I cannot understand the first sentence of Section 2.2, after read it many times.
What is the exact necessary and sufficient optimality condition for the coupling function being convex-concave and differentiable?
Before Equation (2), the authors didn't explain what is monotone and what is monotone inclusion.
I don't think these concepts are very famous in machine learning community.

In Section 2.3,
the authors provided the introduction of lots of iterative methods.
It is difficult for me to distinguish which are completely new findings for solving their inclusion problem and which are the existing results.
The authors completed a literature review here.

In Section 2.4,
I don't know why the problem (1) can be written as Equation (8).
Could you provide more explanations about this point?
Limited space of one submission should be on important points.

In Section 3,
could you explain more about why $G_B(w)$ in Equation (10) is defined as this form?
Just for facilitating the proof of Theorem 3.1 and Theorem 3.2?

In Section 3.2,
the authors provided a generalized FBF algorithm.
Isn't this Algorithm 3.1 a combined re-written version of Equation (4) and Equation (5)?

There is a big gap between Equation (9), Theorem 3.1 and Theorem 3.2 and the experimental results shown in Section 4.
Besides, there are no open source codes provided.
It is very hard for me to figure out the details of the experiments and meantime to check the reproducibility of this paper.

Although, the authors stated that "Due to the theoretical nature of this work, the aim of this section is not to achieve new state-of-the-art
results."
I don't think optimization is a theoretical branch of our machine learning community.
If a proposed optimization method cannot be proved to be very useful in certain ares or specific tasks,
it will be very doubting.
If we intend to do theoretical contributions,
we should try to prove the theoretical properties or convergence bounds for the existing useful optimization methods.

Since ICLR is a highly selective conference,
the originality and significance of one submission will always be in the first priority.
I cannot accept this paper in current state.

---

> ### Author Response · Authors · 2020-11-13
> **Response to Reviewer1**
>
> Dear reviewer, thank you for your interest and the comments. For your convenience we highlighted all the changes in the pdf in red.
>
> > Why do you introduce h(y) here?
>
> The functions $h$ and $f$ here act as regularizers. These could reflect the box constraints in the case of weight clipping or the spectral normalization in the case of SN-GANs. We added an explanatory remark in Section 2.4 of the manuscript. These functions are mentioned explicitly and treated separately in the algorithms because of their potential nonsmoothness.
>
> > Later the author restrict Equation(1) to a deterministic version
>
> We restrict to the deterministic version only in the introductory Section 2. We try to differentiate between the stochastic and the deterministic setting not via the expectation in the objective function but whether stochastic or batch gradient updates are used. This was done purely for ease of notation in order to not having to write the expected value whenever the objective function appears. We tried to clarify this in the manuscript right below eqn (1).
>
> > I cannot understand the first sentence of Section 2.2.
> > I don't know why the problem (1) can be written as Equation (8)
>
> What we meant was that the monotone inclusion corresponds to the first order optimality condition of our main problem.
> In general finding a first order critical point stipulates a necessary condition for being a local solution. Due to the assumed convexity of the problem every critical point is indeed a solution.
> Thus, solving the (monotone) inclusion is equivalent to solving the minimax problem and finding a saddle point. We clarified this point in the updated version.
>
> > the authors provided [...] lots of iterative methods. It is difficult for me to distinguish
> which are completely new
>
> We rephrased this section slightly in order to highlight the origin of the different methods.
>
> > could you explain more about why $G_B (w)$ in Equation (10) is defined as this form?
>
> We added a game theoretic interpretation of the minimax gap. In short it corresponds to the payoff each player can achieve by choosing the best response given the (suboptimal) strategy of their opponent. A set of strategies thus corresponds to a small gap if only a small payoff can be achieved for either player by playing the best strategy given the current one.
>
> > In Section 3.2, the authors provided a generalized FBF algorithm. Isn't this Algorithm 3.1 a combined re-written
> version of Equation (4) and Equation (5)?
>
> Indeed. It is a template in order to reconstruct these two methods and thus highlights how they are connected.
>
> > There is a big gap between Equation (9) [...] and the experimental results
>
> In Section 2.2 we tried to make the connection between monotone inclusions and minimax problems clear. If it helps we can explicitly write down our methods for saddle point problems (which includes the training of GANs) in the Appendix.
>
> > there are no open source codes provided
>
> These should be present in the supplementary material. If there is a problem with the zip file or the download, please let us know.
>
> > Although, the authors stated that "Due to the theoretical nature of this work [...]"
>
> We rephrased the beginning of the experiment section in order to highlight our practical contributions.

---

### Decision · Program_Chairs · 2021-01-07
**Final Decision**

**Decision:**

Reject

**Comment:**

This paper provides a unified view of some known methods for monotone operator inclusion problems like Forward-Backward-Forward (FBF) and OGDA, and provides new convergence results for the stochastic version of a variant of FBF called FBFp. All reviewers initially recommended rejection. The rebuttal and the manuscript update addressed several concerns from the reviewers, though the general consensus after rebuttal was still that the paper lacked in significance for the ICLR community. The AC thinks that the paper could make an interesting overview paper in a more optimization / theoretically minded venue.